# The Determinant of *Sukuk* Rating: Agency Theory and Asymmetry Theory Perspectives

Bedjo Santoso [1,*], Widodo Widodo [1], Muhammad Taufiq Akbar [2], Khaliq Ahmad [3] and Rahmat Heru Setianto [4]

1   Department of Management, Faculty of Economics, Universitas Islam Sultan Agung, Semarang 50145, Indonesia; widodo@unissula.ac.id
2   Directorate General of Taxes, Indonesian Finance Ministry, Semarang Branch Office, Semarang 50188, Indonesia; muhammad.taufiqakbar@pajak.go.id
3   Putra Business School (PBS), Universiti Putra Malaysia, Serdang 43400, Malaysia; khaliq@putrabs.edu.my
4   Department of Management, Universitas Airlangga, Surabaya 60286, Indonesia; rahmat.heru@feb.unair.ac.id
*   Correspondence: bedjo.s@unissula.ac.id

**Abstract:** This research aims to develop a determinant variable of the *Sukuk* rating derived from agency and asymmetry theories. This research is essential because *Sukuk* or Islamic Bonds is needed in Indonesia, with 85% of its population out of 320 million people being Muslim. Many studies on the determinants of *Sukuk* ratings have been conducted and are still trending research. However, they are rarely observed from the perspective of agency and asymmetry theories, which are the basis for the relationship between principals and investors. The relationship produces three primary variables in the *Sukuk* rating determinants, namely financial disclosure quality (FDQ), accounting-based risks (ABRs), and earnings management (EM). This research used 570 panel annual reports from 2018 to 2020 and involved 190 firm-issued *Sukuk*. Meanwhile, the variables' reflection used several indicators. SEM (structural equation modeling) was used for the statistical analysis with the help of PLS—primarily smart PLS version. The results exposed that FDQ, ABRs, and EM derived from the two theories are affected significantly by the determinant of the *Sukuk* rating. In comparison, earnings management successfully moderates the FDQ and *Sukuk* rating variables but fails to moderate the ABRs to the *Sukuk* rating. The conclusion also revealed that these relationship theories are fundamental in developing the *Sukuk* rating. However, the variables should be more complex for future research. With significant results, the agency and asymmetry theories proxied by three variables can explain the *Sukuk* rating. Accordingly, these theories are relevant as approaches in determining important factors of the *Sukuk* rating.

**Keywords:** *Sukuk* rating; agency theory; asymmetry theory; financial disclosure quality; accounting-based risks; earnings management





## 1. Introduction

The *Sukuk* rating is crucial, especially in Indonesia, which aggressively promotes *Sukuk* investment for infrastructure development. The issues according to *Sukuk* investment are primarily related to the success of the problems in integrity, accountability, transparency, and trust of a country to convince investors. Indonesia is still low in terms of good corporate governance (GCG), as seen in the GCG rating. It ranked 57 globally and placed fourth in ASEAN (World Economic Forum 2019) according to those indicators. The study of the *Sukuk* rating cannot be separated from those aspects, especially in the field of accountability, integrity, and transparency of state financial management.

The majority of previous studies related to *Sukuk* ratings focused mainly on the relationship between *Sukuk* ratings and leverage as researched by Elhaj et al. (2015, 2017). In comparison, some research was also conducted by Al-Homsi et al. (2017), Ulfa (2019), Toumeh et al. (2020), Yuwono and Aurelia (2021), and Ningrum et al. (2019).

Many previous studies on *Sukuk* (Islamic bonds) have been carried out, but not as much on bonds. The determinants of *Sukuk* are determined mainly by the firm's endogenous factors. Meanwhile, in addition to endogenous factors, bonds are also determined by interest rate as the exogenous factors. According to Zhou (2021), bond yields, which are an essential determinant in the bond rating, are strongly influenced by short-term interest rates, inflation rates, and economic growth. However, in this study, the interest rate is excluded because *Sukuk* and bonds have different values. *Sukuk* does not accept interest rates because there is no term interest in Islam. Furthermore, Dang and Huynh (2020) postulated that bank credit risk influences people to invest in government bonds, whereas credit and bonds are the most determinants to support economic growth.

Some studies connected *Sukuk* ratings with transparency, accountability, and earnings management. They are, however, still partial, studies conducted by Hassan and Marston (2019), and Al-Homsi et al. (2017). They only discussed the relationship between *Sukuk* rating on leverage and current ratio variables. Agustia and Suryani (2018) demonstrated the relationship among financial disclosure quality (FDQ), leverage ratio, and interest coverage on the *Sukuk* rating, and focused on return on asset, *Sukuk* structures, and the *Sukuk* rating. In comparison, Prastiani (2018), Ulfa (2019), Hassan and Marston (2019) observed the role of return on investment (ROI), FDQ on bond rating. Qizam and Fong (2019) emphasized FDQ, ABRs, and ROI on *Sukuk* ratings. Arundina et al. (2016) examined total assets, leverage ratio, and current ratio on the *Sukuk* rating.

When discussing the *Sukuk* rating, one needs to explore the asymmetry theory approach that describes the relationship between investors and *Sukuk* issuing. Both parties have different information; one party has more information than another. For example, the firm's management has more information than investors about the capital market. The level of this information asymmetry varies from very high to very low. Information asymmetry has a real effect on finance and financial decisions Elements of asymmetry theory include trust, moral hazard, reputation, and opportunistic behavior.

Furthermore, accounting and financial data are expected to minimize conflicts of interest between parties obsessed with interest in the firm (Jensen and Meckling 1976; Watts and Zimmerman 1986). Nonetheless, it is generally accepted that accounting and financial data can cause management discretion and is expected to reduce the gap between the relationship of the principal and agent where information asymmetry or information imbalance (asymmetrical information) prevails.

Additionally, the *Sukuk* rating needs to be studied from theoretical agency that describes the relationship or contract between the principal and the agent. The principal employs the agent to perform tasks for the principal's interest, including delegating decision-making authority from the principal to the agent (Melzatiaa et al. 2019). Agency theory is appropriate to apply to *Sukuk* issues, including investment, *Sukuk* issuance, and *Sukuk* rating. It is similar to the relationship between the principal and agent in agency theory. Agency theory contains many elements, including risk performance, moral hazard, information asymmetry, incentives, cost, and monitoring. The decrease in the theoretical model resulted in the FDQ and ABRs and earnings management variables. The model refers to the asymmetry and agency theories, which affect these three variables and the *Sukuk* rating. The theoretical derivation of the model will be explained in Section 2.

According to Qizam and Fong (2019), research on *Sukuk* ratings rarely focuses on aspects of FDQ and ABRs. He added that the ABRs variable in the *Sukuk* rating was still floating. Unfortunately, these studies mostly adopted short-term measures of FDQ (e.g., information content, accrual quality, or other measures). They did not utilize a large potential of long-term financial information.

Research on the relationship between FDQ and *Sukuk* ratings is still mostly focused on the short-term financial information quality in the capital market, the downtrends of *Sukuk* ratings, and bond ratings amid the rapid growth of *Sukuk* presumably related to the FDQ. Thus, future research needs to develop the determinants of disclosure quality that involve important attributes of financial disclosure and fundamental or ABRs. Further research is

needed involving FDQ, ADQ (accounting disclosure quality) and earnings management variables. Previous research shows an inconsistent/unclear relationship between FDQ and *Sukuk* ratings, i.e., Qizam and Fong (2019) and Al-Homsi et al. (2017) stated that financial reporting quality has a significant negative effect on *Sukuk* ratings.

Meanwhile, the relationship between ABRs and *Sukuk* ratings is also inconsistent, as reflected in the following studies: Qizam and Fong (2019), Al-Homsi et al. (2017), Arundina et al. (2016), Ningrum et al. (2019). Each explained that ABRs had an effect on the bank's rating.

Further research by Ningrum et al. (2019), Prastiani (2018), and Ulfa (2019) shows that there is an indefinite relationship between earnings management and *Sukuk* ratings where inconsistencies are still found in the relationship between earnings management and *Sukuk* ratings.

Additionally, the imprecise relationship between FDQ and earnings management is shown by Hassan and Marston (2019), who assert that if the FDQ is low, the tendency to make earnings management adjustments is higher. In addition, the relationship between the ABR variable and earnings management is indistinct, as found in research by (Yuwono and Aurelia 2021), Astuti (2020), Ningrum et al. (2019), and Ulfa (2019). They concluded that financial leverage as an indicator of ABRs had an effect on earnings management.

Motivated by these previous studies and research gaps, this research intended to examine the determinants of FDQ on employing proxies of the relevance of ratio (represented by the CMH variable). It reflects the covariant of intrinsic value and market value divided by intrinsic value and book value. Meanwhile, the reliability of the ratio (represented by the VMH variable) demonstrates a variant of market value divided by a variant of book value on the *Sukuk* rating. Furthermore, this research is aimed to observe ABRs proxied by total debt ratio (total debt/equity) on the *Sukuk* rating. This research also examines earnings management proxied by the modified Jones model on the *Sukuk* rating.

This research was conducted in Indonesia because the growth of *Sukuk* in Indonesia has been increasing from year to year, especially after the Financial Services Authority Regulation (POJK) No. 18/POJK.04/2015 concerning *Sukuk* Issuance and Requirements.

According sharia capital market statistics—2021 issued by *The Financial Services Authority* (*OJK–Otoritas Jasa Keuangan*) the number of *Sukuk* issuances increased after the 2015 Financial Services Authority Regulation (https://www.ojk.go.id/id/kanal/syariah/data-dan-statistik/data-produk-obligasi-syariah/Default.aspx, accessed on 28 February 2022). In 2015, there were 47 outstanding *Sukuk* with a total value of IDR 9.9 billion. In 2016, there were 53 outstanding *Sukuk* with a total value of IDR 11.8 billion. Then, up to January 2020, the total issuance of *Sukuk* was the highest with a total of 142 *Sukuk* and still outstanding with a total value of IDR 29.66 billion. However, from 2019 to 2020, the growth was quite stagnant, and based on the same source until July 2021, the growth declined, whereas the requirements of *Sukuk* fund were very high (OJK 2021).

Elhaj et al. (2017) stated that with the increasing growth of significant *Sukuk* from year to year, the issue of *Sukuk* rating became decisive. Furthermore, there are two types of *Sukuk* issued in Indonesia, namely those issued by the government called state *Sukuk* or SBSN (State Sharia Bonds) and those issued by companies called corporate *Sukuk*. According to Astuti (2020), a study on GCG, which consisted of default risk, accountability, disclosure, and transparency, is considered crucial (Pranoto et al. 2017).

Most of a firm's assets in the past were tangible assets, such as buildings and machinery. Now 'magic assets' (the intangible assets), including brands, corporate image or goodwill, intellectual property, and human assets, are increasingly dominant. In addition, the evolution of modern companies, proliferating into conglomerates, subsidiaries, or leased assets, is also increasingly blurring the firm's traditional boundaries.

The development of investment models due to the disruptive evolution of 'magic firms' requires adjustments in the accounting systems. It needs to reveal relevant and reliable financial information. Islamic Financial Institutions (AAOIFI) is the authorized global institution for developing Sharia Standard (Bouheraoua et al. 2014) and IAS 38

(International Accounting Standards Board 2016). In the capital market, most empirical evidence indicates that the quality of financial disclosures, relevance, reliability, and ABRs are highly valued empirically (Botosan and Plumlee 2013; Qizam 2011; etc.). However, most studies related to these variables adopted short-term measures and did not utilize the great potential of long-term financial information.

Meanwhile, the current global economy, especially Islamic finance, is growing rapidly, including *Sukuk*. It is suspected that *Sukuk* is a safe Sharia fund that does not engage in excessive speculation and has a low trading turnover. It makes *Sukuk* less volatile than conventional bonds. Therefore, issuers of *Sukuk* are global and are not only from Islamic countries but also from other Western, African, and Asian countries. Previous issuers of conventional instruments were the UK, South Africa, Luxembourg, and Hong Kong. Currently, the volatility of *Sukuk* developments is primarily due to the volatility of oil prices. It results in a decline in revenues of USD 300 billion, pushing the budget deficits of the Gulf Cooperation Council (GCC) countries, offering future opportunities and new challenges for *Sukuk* growth. Consequently, most GCC countries tried to switch to the capital market through the issuance of bonds and *Sukuk*.

Optimistic future developments for *Sukuk* growth have an average of 10% per year. Additionally, it has a positive gap between supply and demand for *Sukuk* in upcoming years, i.e., USD 143 billion (2017), USD 178.4 billion (2018), USD 221.1 billion (2019), USD 256.9 billion (2020), and USD 271.3 billion (2021) (Reuters 2017). Therefore, it needs the quality of Jolly's financial information to guarantee and support opportunities for a positive response and an optimistic global growth of *Sukuk*/bond rating, to ensure trust, transparency, accountability, and credibility. Hence, every country that wants to develop *Sukuk* requires globally accepted standards and rules.

*Sukuk* is regulated by The Accounting Board (AAB) of the Accounting and Auditing Organization for AAOIFI, which has officially issued the financial accounting standard (FAS). FAS 33 (which supersedes earlier FAS 25) that sets out the improved principles for classification, recognition, measurement, presentation, and disclosure of investment in *Sukuk*, shares, and other similar instruments of investments made by Islamic financial institutions (IFIs/the institutions), in line with Sharia principles. It defines the key types of instruments of Sharia-compliant investments and defines the primary accounting treatments commensurate to the characteristics and business model of the institution under which the investments are made, managed, and held. FAS 34 aims to establish the principles of accounting and financial reporting for assets and businesses underlying the *Sukuk* to ensure transparent and fair reporting to all relevant stakeholders, particularly including *Sukuk* holders.

In Indonesia, *Sukuk* products refer to the Statement of Financial Accounting Standards (PSAK). PSAK 110 was established in 2011 (and revised in 2015). These accounting standards refer to the IFRS (International Financial Reporting Standard) as the reference standard for financial accounting in Indonesia. The government expects the comparability of financial reports between countries to be recognized globally and to be of higher quality through internationally accepted standards.

It is crucial to ensure efficiency in a global competition that requires reliability and relevancy (Qizam and Fong 2019). Moreover, in the globalization era, every country that wants to be an international business player must pay attention to the above-mentioned issues. It is also essential for policymakers to determine the role of institutions, policy instruments, and factors, which are necessary for attaining higher productivity, efficiency, and profitability and for better withstanding forces of competition on global and regional markets (Zeibote et al. 2019).

## 2. Literature Review and Hypotheses Development

This section discusses theoretical perspectives related to *Sukuk* to acquire the basic theoretical model and empirical research model. First, it will discuss the importance of the asymmetry, agency, and signaling theories in *Sukuk* research (rating). These three theories

are very relevant since they are a theoretical novelty in *Sukuk* research. Furthermore, the issues regarding *Sukuk* (theories) are discussed and then a basic theoretical model is developed to obtain an empirical model. Studies in previous research are explained to formulate research hypotheses.

### 2.1. Agency, Asymmetry, and Signaling Theories and Highlight Perspective

As stated in the introduction, agency theory, asymmetry theory, and signaling theory have a relationship with *Sukuk* ratings. Hence, it is necessary to develop a model that describes the relationship among them. The *Sukuk* rating is considered crucial for investors because the information is used to make investment decisions. Factors that affect the *Sukuk* rating are financial and non-financial. Financial factors include solvency, leverage, and earning ability. Meanwhile, nonfinancial factors are aspects of GCG, including trust, accountability, transparency, moral hazard, signal information, signal force, and benefits, which lead to information asymmetry that impacts perception.

### 2.2. Asymmetry Theory

According to Borhan and Ahmad (2018), *Sukuk* ratings can assist investors in making the right investment decisions. It is because *Sukuk* ratings can reduce information asymmetry between *Sukuk* issuers and investors (Hemraj 2015). The higher the *Sukuk* rating, the lower the risk. *Sukuk* rating is also an indicator of the timeliness of principal and rental payments or *Sukuk* margins (Al-Haraqi and Endang 2017).

Asymmetry theory conveys that the parties related to the firm do not have the same information about the prospects and risk of the firm. In contrast, certain parties have better information than other parties. Information asymmetry occurs where one party has more information than another. The level of this asymmetry information varies from very high to very low. Information asymmetry has a real effect on investment decisions (Atmaja 2008).

Managers usually have better information than outsiders (such as investors). In other words, there is information asymmetry between managers and investors. Investors who suppose they have less information will try to interpret the behavior of managers. This situation will trigger a condition known as information asymmetry. There is an imbalance in the acquired information among management as an information provider, shareholders, and stakeholders in general as information users.

According to Scott (2006), information asymmetry will encourage moral hazard. The holder does not entirely know the activities carried out by the investment principal. Consequently, it will allow actions that are far from the knowledge of each other in such a way that ethically those actions may not be feasible.

The existence of information asymmetry is a source of conflict between principals and agents to try to take advantage of each other for their interests. Lin et al. (2015) described that the measurement proxies of information asymmetry consisting of firm size, firm age, the proportion of shares offered to the public, and the effect on underpricing of the underwriter's and auditor's reputation. Meanwhile, the firm size, age, and the proportion of shares offered do not affect underpricing.

### 2.3. Signaling Theory

Signaling theory discusses the rise and fall of prices in the market; hence, it will influence investor decisions. Investor response to positive and negative signals is greatly affected by market conditions. Investors will react in various ways in response to these signals, such as by chasing sold stocks or taking action in the form of inaction such as "wait and see" or by waiting to see how the progress lasts and then taking action. The "wait and see" decision is not about right or wrong, but it is an investor's reaction to avoid the greater risk due to market factors with no earning ability or nothing favorable to them him (Chandra 2011).

Tamara et al. (2020) expressed that good quality companies will intentionally signal the market. By doing so, potential investors are expected to distinguish between good and

bad quality companies. When making a public offering, potential investors cannot fully differentiate between good and bad quality companies. Therefore, issuers and underwriters deliberately give signals to the market.

Ross (1997) developed a model in which the structure of the model is a signal conveyed by managers to the market. If the manager has confidence that the firm's prospects are good and wants the stock price to increase, they will communicate this to investors. The signaling theory reveals the encouragement of companies to provide financial statement information to external parties because of the information asymmetry between the firm and external parties (investors and creditors). As a result, management has more information and prospects for the firm in the future (Fadah et al. 2020).

However, sometimes management provides information on the firm's condition that is not under the actual conditions. Consequently, investors find it difficult to distinguish between companies of high quality and those of low quality (West 1970). Companies can increase the firm value by reducing information asymmetry, namely by giving signals to outsiders (Tamara et al. 2020).

A *Sukuk* rating reflects the firm's performance, providing a positive signal that can influence investors' opinions to make investment decisions. In signal theory, a high *Sukuk* rating shows a positive signal about the future condition of the firm (Mariana 2016). In addition, the way to reduce information asymmetry is to provide signals to outsiders that contain transparent, credible, and quality information as disclosed by the firm in the financial statements (Fadah et al. 2020).

*2.4. Agency Theory*

Agency theory is concerned with resolving two problems in principal–agent relationships. The first issue arises when the two parties have conflicting objectives. It is difficult for the principal to verify what the agent is doing and whether it has behaved appropriately. The second issue is the risk sharing that occurs when the principal and the agent have different attitudes to risk (due to various uncertainties). Each party may prefer other actions because of their different risk preferences.

The concept of agency theory is the relationship or contract between the principal and the agent. Jensen and Meckling (1976) illustrated an agency relationship as a contract under one or more persons (the principal(s)) engaging with another person (the agent) to perform some service on their behalf. It involves some decision-making authority to the agents, and the principal employs the agent to perform tasks for the principal's benefit. Companies with shareholders act as principals and management as their agents. Shareholders utilize management to act in the interests of shareholders.

Agency theory assumes that all individuals act according to their interests. Eisenhardt (1989) framed that people, in general, will work opportunistically by prioritizing their interests both as agents and as principals, each of which will optimize their respective benefits. Thus, the firm has two different interests, each striving to achieve the desired prosperity. It results in a possibility of a conflict of interest between the agent and the principal. It is what is called the agency problem related to moral hazard. The conflicts of interest in the agency issues are the principal (*Sukuk* provider) and agent (*Sukuk* investor).

Borhan and Ahmad (2018) pointed out that conflicts of interest between the principal and the majority agent are between shareholders and firm management and between shareholders and debt securities holders or creditors. Borhan and Ahmad (2018) explained that bondholders are creditors with a different perspective on how the firm should be managed compared to shareholders.

Bondholders usually choose carefully to manage companies to pay the debts on time. At the same time, shareholders will usually demand the firm to be more aggressive in investment about the expansion and growth of the firm. Borhan and Ahmad (2018) mentioned that aggressive investment could lower bond ratings. Bondholders may suffer losses by owning low-rated bonds. Debt agreements can resolve conflicts of interest

between shareholders and bondholders. It also can protect bondholders from possible defaults and improve credit ratings (Borhan and Ahmad 2018).

There is a lot of interest in agency theory related to Islamic banking and finance, especially regarding the contractual relationship between the *Sukuk* holder (principal) and the firm (agent) in several different types of *Sukuk*. Borhan and Ahmad (2018) justified that agency theory and other related theories can be adopted in conceptualizing the relationship between *Sukuk* ratings and their determinants.

*2.5. Sukuk Rating*

Erdoğan et al. (2018) mentioned three major rating agencies: Standard and Poor's, Moody's, and Fitch. Ratings have dominated 90% of the global market. In addition, the credit rating industry was established approximately 158 years ago. Many types of market participants have also utilized these credit ratings. Credit ratings for *Sukuk* are also used for buying or selling, private contracts, and regulators of the investors (Covitz and Harrison 2003). It aims to assess credit risk, comply with the guidelines or rules of investment, and determine the amount of collateral that withstand credit derivatives exposure.

However, many contenders criticized the methodology and other issues. These issues should be developed to deal with the currently growing financial system. In addition, the informational effects of rating changes have become more pronounced following the implementation of the regulation. Jorion and Zhang (2007) suggested the growing importance of credit ratings to capital and bond market participants.

*Sukuk* ratings and bond ratings are crucial for issuers and investors to measure the probability of debt failure and the firm's risk, which functions as the issuer. It can be concluded that *Sukuk* ratings and bond ratings are used to measure the level of risk and return on investment in a firm. The better the *Sukuk* rating and corporate bonds, the lower the risk of default on the *Sukuk* and bonds will be. Thus, this will indirectly become a condition for investing in this long-term capital market instrument (Sudaryanti et al. 2011).

The *Sukuk* rating is a process of assessing the payment of principal and or interest possibility with a predetermined time. The higher the rating of the *Sukuk*, the smaller the risk will be (Elhaj et al. 2015). The *Sukuk* rating describes the prospect of the feasibility of a *Sukuk* being worthy for purchase as one of the firm's current assets. *Sukuk* is purchased based on recommendations from a trusted and tested rating agency. *Sukuk* ratings play an important role in providing information and signaling the probability of a firm's debt default (Situmorang 2017). Meanwhile, the benefits of *Sukuk* ratings for companies (Sudaryanti et al. 2011) include: (1) Obtain information on the firm's business position; (2) Determine bond structure from details on the weaknesses and strengths of the firm's management; (3) Support for firm performance; (4) as a firm marketing tool; (5) Maintain investor confidence.

*2.6. Studies Related to Sukuk*

This research links GCG variables with the agency, signaling, stakeholder asymmetry, and legitimacy theory. The theory is engaging as evidenced by a study conducted by Hermawan and Gunardi (2019). They linked social responsibility disclosure and GCG variables with the theoretical framework of stakeholders. Meanwhile, Czaja-Cieszyńska et al. (2021) correlated the variable of social disclosure with business transparency, which is a crucial element in GCG. Therefore, this study will attempt to examine the effect of GCG variables on the *Sukuk* rating.

Studies that discussed FDQ, financial information, GCG, firm size, earning ability, and leverage were conducted by Elhaj et al. (2015), Al-Homsi et al. (2017), Arundina et al. (2016), Qizam (2011), and Farooq and Bari (2015). They suggested that financial leverage is negatively related to financial measures and the *Sukuk* rating relationship. Meanwhile, *Sukuk* ijarah is positively associated with the *Sukuk* structure and *Sukuk* rating relationship. They added that a positive *Sukuk* credit rating is related to financial information, governance

attributes, and *Sukuk* structure. Firm size, earning ability, and leverage have a significant positive effect on *Sukuk* credit rating. Increased disclosure reduces the cost of equity. The risk of information from tradeoff variability between relevance and reliability has a moderating effect on the relationship between ABRs and the cost of equity. Sharia-compliant companies vs. non-Sharia-compliant companies show a different FDQ for the *Sukuk* rating.

Research that involves accounting quality, credit rating, default risk, equity, and disclosure variable were analyzed by Botosan and Plumlee (2013). They found that the decline in credit ratings and tightening of standards affect *Sukuk* ratings, and accounting quality plays an important role in *Sukuk* ratings. Credit rating and default risk have a negative effect on the *Sukuk* rating. The increase in disclosure will reduce the cost of equity and affect the *Sukuk* rating.

Furthermore, studies that investigated financial information, asset, current ratio, interest, return on asset (ROA), return on investment (ROI) and operating income were done by Jolly (2022), Nabila and Arundina (2019), and Qizam and Fong (2019). They concluded that the better the quality of financial information, the more positively the *Sukuk* rating was influenced. Accounting information will affect FDQ and strengthen the *Sukuk* rating. The total asset, long-term leverage ratio, current ratio, ROA, ROI, interest coverage ratio, *Sukuk* structure, and operating income significantly affect the *Sukuk* rating.

Studies by Qizam and Fong (2019) show that FDQ related to reliability affects *Sukuk* ratings, not bond ratings. Leverage is found to be the most influential on *Sukuk* and bond ratings. FDQ is, to some extent, affected the relationship between ABRs (i.e., operating income, leverage, and ROI) and *Sukuk* or bond rating ABRs. Based on the previous research review, researchers on *Sukuk* ratings have the latest trends involving variables of FDQ and ABRs; however, several researchers have inconsistent results in the relationship between these variables and the *Sukuk* rating. The relationship among variables is not determined. Hence, another variable is needed as an intervening one to improve the relationship in the model formed.

### 2.7. Theoretical Model

As mentioned in the discussion, the derivation of the theoretical model in this study is different from the existing model. This study used the asymmetry theory, agency theory, and signaling theory. These three theories are very relevant to be developed in *Sukuk* rating research to form a new theoretical novelty. The theoretical model will produce research variables that will then be built into an empirical model. The two models are shown in Figure 1.

### 2.7.1. The Effect of Financial Disclosure Quality (FDQ) on *Sukuk* Rating

Financial reporting is the presentation of financial statements in the form of numbers in financial reporting. Meanwhile, financial disclosure not only contains reports in numbers but also an analysis in a text to support the firm's financial statements Hassan and Marston 2019). Due to the inconsistency of previous study results, it is still interesting to study further.

Information asymmetry occurs when one party has more information than the other party. In this case, investors and managers allow information asymmetry since managers have more information than investors. Therefore, the role of quality financial reporting/disclosure and *Sukuk* rating can reduce information asymmetry. The research conducted by Qizam and Fong (2019) revealed that FDQ related to the reliability of financial statements causes a positive and significant effect on the *Sukuk* rating. On the other hand, research conducted by Al-Homsi et al. (2017) justified that financial reporting quality as measured by timeliness of reporting has a significant negative effect on the *Sukuk* rating.

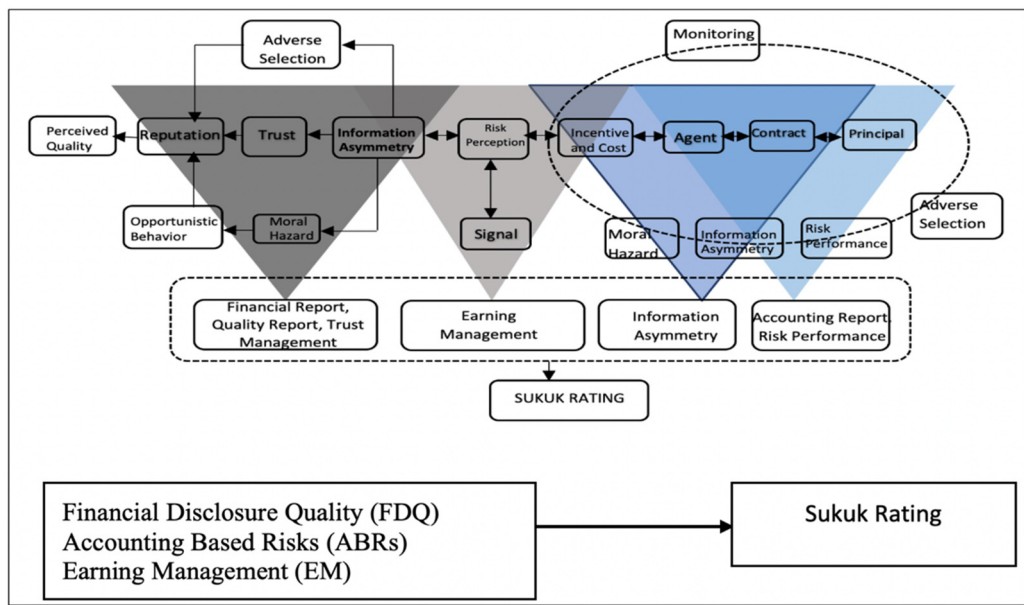

**Figure 1.** Theoretical Models; the Relationship among *Sukuk* Ratings and Asymmetry, Agency, and Signaling Theories.

The FDQ indicator used in this study refers to the research conducted by Qizam and Fong (2019), namely the level of relevance and reliability of the FDQ. The level of relevance is measured by using the ratio between intrinsic and market values divided by intrinsic and firm book value. Meanwhile, the reliability of FDQ is measured using the ratio between market value divided by the book value of the firm in a period. Based on the previously explained description, the research hypothesis can be formulated as follows:

**Hypothesis 1 (H1).** *FDQ has a significant positive effect on Sukuk rating.*

2.7.2. The Effect of Accounting-Based Risks (ABRs) on *Sukuk* Rating

ABRs or accounting ratio-based risks are one of the measures used by rating agencies in assessing a firm's creditworthiness (Al-Homsi et al. 2017). ABRs support investors in decision making, not only in the stock market but also in the debt market.

Financial leverage is a ratio used to assess how much of a firm's assets are financed using debt (Agustia and Suryani 2018). The higher the firm's leverage ratio is, the higher the risk of the firm paying its obligations, impacting creditor confidence, and lowering the *Sukuk* rating. A high leverage ratio value indicates that the firm has a lot of debt to external parties, and there is a possibility of default.

The ABRs indicator used in this study refers to the research conducted by Qizam and Fong (2019), namely the total debt ratio that compares total debt with total equity. Based on the description above, the research hypothesis can be formulated as follows:

**Hypothesis 2 (H2).** *ABRs have a significant negative effect on Sukuk rating.*

2.7.3. The Effect of Earnings Management on *Sukuk* Rating

Watts and Zimmerman (1986) exclaimed that earnings management occurs when managers have discretionary behavior related to accounting numbers with or without restrictions, and this behavior can be adopted to maximize firm value. Earnings management makes the firm's performance look suitable to investors by increasing the firm's earnings (Oktaviyani 2021). Prastiani (2018) stated that earnings management and bond issuance could influence a firm's bond rating. It is by increasing firm earnings so that the firm's performance looks good in the eyes of investors, and investors ultimately entrust them to provide debt to the firm. The firm's management tends to carry out earnings

management or earnings engineering in the period around the bond issuance to make the firm's performance look good. It will impact obtaining bond ratings to increase the firm's attractiveness in the eyes of creditors. The earnings management indicator in this study refers to the research recommendations of Ningrum et al. (2019), which is measured by using the modified Jones model. Based on the description above, the research hypothesis can be formulated as follows:

**Hypothesis 3 (H3).** *Earnings Management has a significant negative effect on Sukuk Rating.*

2.7.4. The Effect of Financial Disclosure Quality (FDQ) on Earnings Management

Hassan and Marston (2019) stated that corporate financial disclosure is any financial information released by the firm, both quantitative and qualitative. The financial information must be reported voluntarily, both formally and informally. Corporate financial disclosure is crucial because it is the primary communication between management and investors in the capital market. According to Alzoubi (2016), financial statements presented by the firm are required to provide more value to its users. The existing consequences of the financial statements presented on the firm's activities encourage managers to adjust the reported financial statements. If the FDQ is low, the tendency of management to make adjustments is higher. On the other hand, if the FDQ is high, management tends not to adjust the reported financial statements.

**Hypothesis 4 (H4).** *FDQ has a significant negative effect on earnings management.*

2.7.5. The Effect of Accounting-Based Risks (ABRs) on Earnings Management

Toumeh et al. (2020) explained that in the case of a high leverage ratio of a firm, the possibility to carry out earnings management is substantial, and the firm has a greater obligation to disclose to the public. When management is motivated to improve its performance, it requires additional working capital from outside parties (loans), which is greater as well. The higher the leverage ratio is, the higher the firm's risk in paying its obligations, which will impact creditor confidence. A high leverage ratio value is considered to have a lot of debt to external parties. These conditions encourage firm management to practice income smoothing (Yuwono and Aurelia 2021). Ningrum et al. (2019) found that financial leverage does not affect earnings management. The results of this study are in line with the results of the research conducted by Ulfa (2019). Based on the description above, the research hypothesis can be developed as follows:

**Hypothesis 5 (H5).** *ABRs have a significant positive effect on earnings management.*

A theoretical framework model was developed according to the formulated hypotheses, as shown in Figure 1. It explains the relationship among variables. The variables in this study consist of independent variables, namely FDQ and ABRs. Meanwhile, the dependent variable is the *Sukuk* rating (SR), and the intervening variable is earnings management (EM). The determination of the EM variable as an intervening variable is to cover the research gap found in previous studies on the relationship among FDQ, ABRs, and SR, with inconsistent results (still debatable/pros and cons). Hence, in this study, the researchers will try to add the EM variable as an intervening variable shown in Figure 2.

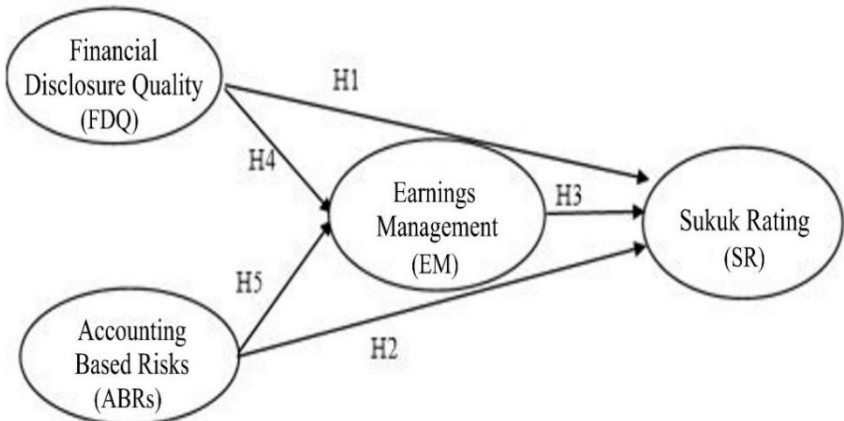

**Figure 2.** Theoretical Framework: Determinant of *Sukuk* Rating with Earnings Management as an Intervening Variable.

### 3. Research Methods

*Sukuk*

The data source in this study were secondary data from the Indonesian Stock Exchange for the periods of 2018, 2019, and 2020. The data used were related to indicators of FDQ, ABRs, earnings management, and *Sukuk* ratings. The sampling technique applied in this study was purposive, nonrandom sampling. Sampling method used purposive sampling, namely *Sukuk* issued by *PT PEFINDO*; the number was 190 (as a cross-section) and combined by three consecutive years (time series) so it was considered a data panel. The number of firms was 190; this excluded 63 firms which were not rated by *PT PEFINDO* (Indonesian Rating *Sukuk* Agency) and 21 firms that had incomplete financial statements for three years (consecutively). Hence, the total data processed were 570 (190 × 3). The detailed descriptions are shown in Table 1.

**Table 1.** Sample Description.

| No. | Criteria | Quantity |
|---|---|---|
| 1 | The number of corporate *Sukuk* that were still listed until 2020. | 274 |
| 2 | The number of Corporate *Sukuk* which were not rated by *PT. PEFINDO* (Indonesian Rating *Sukuk* Agency) for three consecutive years. | 63 |
| 3 | The number of companies that published incomplete financial statements for three consecutive years. | 21 |
| 4 | The amount of data that can be used. | 190 |

Source: Indonesia Stock Exchange.

This study employed the quantitative method structural equation modeling (SEM) by using smart partial least square (Smart-PLS) version 3.3.2. According to (Wold et al. 1996), PLS is a powerful cause analysis method not based on many assumptions or conditions, such as normality and multicollinearity tests. This method has its advantages: the data do not necessarily have a multivariate normal distribution. The indicators with categorical, ordinal, interval-to-ratio data scales can be used. Another advantage is that the sample size does not have to be significant.

The PLS approach is distribution-free, and, therefore, PLS was appropriate for this research because the variable indicators were ratio or percent, nominal, and ordinal. PLS can also analyze constructs formed with reflective and formative indicators (Ghozali 2014). PLS uses a 3-stage iteration process. The first stage produces the weight estimation; the second stage produces an inner model and outer model estimation, and the third stage

produces means and location estimation (Ghozali 2014). As mentioned previously, this study employed three variables, and some indicators are as follows:

$$\text{TAcc}_{it} = \text{EBIT}_{it} - \text{CFO}_{it} \qquad (1)$$

$$\frac{\text{Tacc}_{it}}{\text{TA}_{it-1}} = \alpha \frac{1}{\text{TA}_{it-1}} = \beta \frac{\Delta\text{Sales}_{it}}{\text{TA}_{it-1}} + Y\frac{\text{PPE}_{it}}{\text{TA}_{it-1}} + \delta\frac{\text{ROA}_{it-1}}{\text{TA}_{it-1}} + \varepsilon_{it} \qquad (2)$$

Tacc = Total accruals; EBIT = Earnings before taxes and interest; CFO = Operating cash flows; TA = Total asset; REV = Revenue; REC = Receivable (net); PPE = Property, plant, and equipment (gross); $\varepsilon$ = Error; ROA denotes net income and $\varepsilon$ is discretionary accruals. The details of the variables employed in this study are presented in Table 2.

**Table 2.** Variable and Operational Definition.

| No. | Variables | Indicators | Sources |
|---|---|---|---|
| 1 | *Sukuk* Rating | Scale 1–18 | (Elhaj et al. 2017; Arundina et al. 2016) |
| 2 | FDQ | The relevance ratio (represented CMH) reflects the covariance of intrinsic value and market value divided by intrinsic and book values. The reliability of the ratio (represented VMH) reflects the market value variance divided by the book value variance. | (Qizam and Fong 2019) |
| 3 | ABRs | Proxy by leverage * Total debt ratio (total debt/total equity) | (Elhaj et al. 2017; Arundina et al. 2016; Qizam and Fong 2019; Faozi and Ghoniyah 2019) |
| 4 | Earnings Management | Modified Jones model ** | (Ningrum et al. 2019; Veganzones et al. 2021) |

Variables Description: * ABRs: Firm risk is measured using leverage, because the higher the level of debt (uncertain level) the higher the firm risk according to capital structure theory (Faozi and Ghoniyah 2019). From accounting perspective, firm risk can be perceived from materiality aspect. In other words, the amount of debt (leverage) materially affects firm risk (firm value). ** Earning management (EM) is earnings manipulation can be addressed by detecting accounting abnormalities (misleading behavior). EM can be proxy by determining the discretionary accrual or abnormal accrual in the company using financial information. EM models have been developed to estimate discretionary accruals and provide evidence that the modified version of the Jones model is the most powerful one to capture the level of earnings management, as follows (Veganzones et al. 2021).

## 4. Results

The results of the descriptive statistical test based on Table 3 show that the *Sukuk* rating variable (RAT) with a sample of 570 has a minimum value of 1, while the maximum value is 18. The mean value of the *Sukuk* rating variable is 14.482, and the standard deviation is 5.668. The ABRs variable, as indicated by the leverage (LEV) indicator through the calculation of the debt-to-equity Ratio (DER), has a minimum value of −2.128 and a maximum value of 7.341 with an average value (mean) of 1.815 and a standard deviation obtained of 2.350. Based on these results, the average *Sukuk* issuing firm in this study tends to have a larger debt composition than its equity. In other words, the potential for default tends to be high.

The earnings management variable (EM), calculated using the modified Jones model, has a minimum value of −0.177 and a maximum value of 0.636 with an average value (mean) of 0.019 and a standard deviation of 0.009. In conclusion, the average *Sukuk* of issuing companies in this study performs earnings management by increasing earnings. However, the average level of earnings management is relatively low, with a value of 0.019. The FDQ variable is calculated by using the indicator of financial statement characteristics. They are in the form of relevance and reliability. The relevance is estimated by comparing the firm's intrinsic value and market value divided by the firm's intrinsic value and book value. Meanwhile, the reliability is summed by comparing the market value with firm book value. The minimum value of relevance is −0.673 with the maximum value of 1.968, an average value (mean) of 0.956, and a standard deviation of 0.469. Meanwhile, the minimum reliability value is −1.927 with a maximum value of 2.541, an average (mean)

of 0.826, and a standard deviation of 0.877. Based on these results, the average companies of *Sukuk* issuers in this study tend to have a high level of FDQ. It means the disclosure of financial statements is relatively complete, and the quality of the information disclosed is fairly good.

**Table 3.** Descriptive Analysis.

| Delimiter: | | | Encoding: | | U1 | |
|---|---|---|---|---|---|---|
| Value Quote Character: | | None | Sample Size: | | 570 | |
| Number Format: | | Europe (example: 1.000, 23) | Indicators: | | 5 | |
| Missing Value Marker: | | None | Missing Values: | | 0 | |
| Indicators | | | Indicator Correlations | | | Raw File |
| | No. | Missing | Mean | Median | Min. | Max. | Standard Deviation |
| RAT | 1 | 0 | 14.482 | 14.000 | 1.000 | 18.000 | 5.668 |
| LEV | 2 | 0 | 1.815 | 1.319 | −2.128 | 7.341 | 2.360 |
| EM | 3 | 0 | 0.019 | −0.041 | −0.177 | 0.636 | 0.009 |
| CMH | 4 | 0 | 0.956 | 0.970 | −0.673 | 1.968 | 0.469 |
| VMH | 5 | 0 | 0.826 | 0.911 | −1.927 | 2.541 | 0.877 |

*4.1. SEM Analysis—Partial Least Square*

The data analysis in the study used the Smart PLS 3.3.2 software program to facilitate data processing. Data analysis using partial least square begins with a validity test (convergent and discriminant validity) and reliability. Evaluation of convergent validity from individual item reliability checks can be seen from the standardized loading factor value. The standardized loading factor describes the correlation between each measurement item (indicator) and its construct. The value of the loading factor is 0.7 and ideal. It means that the indicator is valid for measuring the construct. Siswoyo (2016) exclaimed that the loading factor value of 0.5 is still acceptable. The test results are as follows:

In Figure 3, the outer loading factor value for all indicators is more significant than 0.70 except for the relevance of the FDQ. It has an outer loading factor value of 0.658 but >0.5; thus, it is still acceptable.

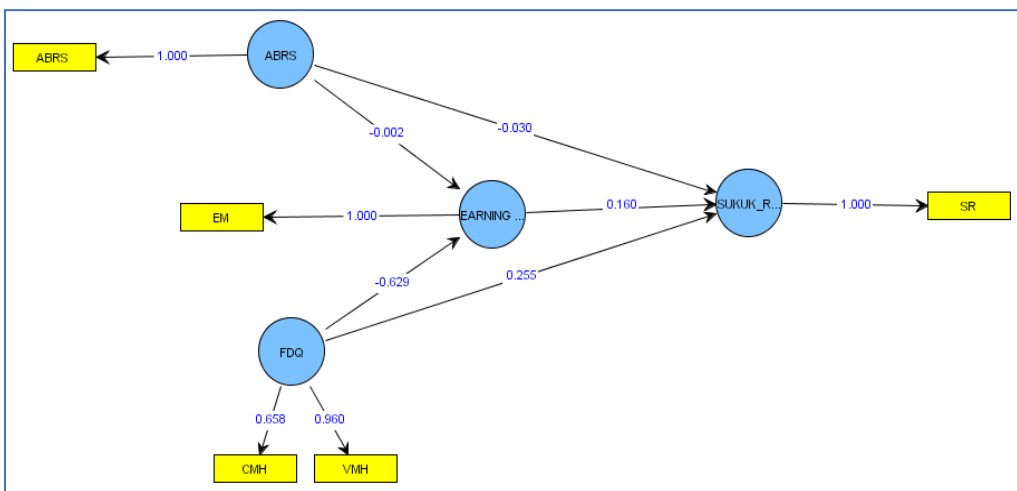

**Figure 3.** Outer and Inner Model.

## 4.2. Internal Consistency or Construct Reliability

Internal consistency or construct reliability is found by observing the value of Cronbach's alpha and composite reliability (CR) Fornell and Larcker (1981), Yamin and Kurniawan (2011), and Siswoyo (2016). The composite reliability (CR) interpretation is the same as Cronbach's alpha. The limit value of 0.7 is acceptable, and 0.8 is very satisfactory. The test results are as follows:

From Table 4, the Cronbach's alpha value of all variables is >0.70, except for the FDQ variable with a Cronbach's alpha value of 0.587. The composite reliability value is >0.70. Hence, the indicator is still acceptable.

**Table 4.** Construct Reliability.

|  | Cronbach's Alpha | rho_A | Composite Reliability | Average Variance Extracted |
|---|---|---|---|---|
| ABRs | 1.000 | 1.000 | 1.000 | 1.000 |
| FDQ | 0.587 | 0.969 | 0.832 | 0.687 |
| Earnings Management | 1.000 | 1.000 | 1.000 | 1.000 |
| *Sukuk* Rating | 1.000 | 1.000 | 1.000 | 1.000 |

Source: Output of SmartPLS 3.3.2.

## 4.3. Average Variance Extracted (AVE)

Another measure of convergent validity is the average variance extracted (AVE) value. The AVE value describes the variance or diversity of the manifest variables that the latent construct can own. Thus, the greater the variance or diversity of the manifest variables is that the latent construct can contain, the greater the representation of the manifest variable will be on the latent construct.

Yamin and Kurniawan (2011), Siswoyo (2016) recommended the use of AVE for a criterion in assessing convergent validity. A minimum AVE value of 0.50 indicates a good measure of convergent validity. The latent variable can explain the average of more than half the variance of the indicators. The test results are as shown in Table 4. The table demonstrates that the AVE value of all indicators is >0.50. Thus, the size of the convergent validity is good.

## 4.4. Discriminant Validity Test

The PLS test is used to find discriminant validity values. Fornell and Larcker procedures can be used by comparing the AVE root with the correlation of the relevant construct to other latent variables. The results of the Fornell and Larcker validity test are as follows (Table 5).

**Table 5.** Discriminant Validity—Fornell and Larcker.

|  | ABRs | FDQ | Earnings Management | *Sukuk* Rating |
|---|---|---|---|---|
| ABRs | 1.000 |  |  |  |
| FDQ | 0.290 | 0.831 |  |  |
| Earnings Management | −0.109 | −0.639 | 1.000 |  |
| *Sukuk* Rating | 0.553 | 0.828 | −0.491 | 1.000 |

Source: Output SmartPLS 3.3.2.

Table 5 presents the results of the AVE root test of 0.831 for FDQ and 1.000 for ABRs, earnings management, and *Sukuk* ratings. Based on the root value of AVE, the correlation of FDQ with the relevant construct is 0.831, which is still more significant than the correlation

of FDQ with other variables (FDQ—leverage is 0.290, FDQ—earnings management is −0.639, and FDQ—*Sukuk* rating is 0.828). Likewise, the AVE root value for the correlation of other variables shows the same result. The construct is valid and reliable based on the reliability and validity tests according to the cross-loading factor and Fornell and Larcker. Thus, it can be continued to test the inner model.

### 4.5. Results of the Inner Model (Structural Model)

The structural model can be evaluated based on the path coefficient parameter and R2 value on the endogenous variables, as shown in Table 6.

**Table 6.** Path Coefficients.

|  | Original Sample | Sample Mean | Standard Deviation | *t*-Statistics | *p*-Values |
|---|---|---|---|---|---|
| ABRs → Earnings Management | 0.509 | 0.205 | 0.297 | 2.812 | 0.038 |
| ABRs → *Sukuk* Rating | −0.344 | 0.311 | 0.125 | 2.819 | 0.027 |
| FDQ → Earnings Management | −0.887 | 0.629 | 0.191 | 2.961 | 0.044 |
| FDQ → *Sukuk* Rating | 0.653 | 0.722 | 0.188 | 3.880 | 0.000 |
| Earnings Management → *Sukuk* Rating | −0.230 | 0.008 | 0.079 | 3.102 | 0.047 |

Source: Output of SmartPLS 3.3.2.

The path coefficient results in the table above show the results of the *t*-test of the influence between exogenous and endogenous variables with a direct path. These results are used to explain hypothesis testing.

The results of the R square value are presented in Table 7 as follows.

**Table 7.** R-Square Results.

|  | R Square ($R^2$) |
|---|---|
| *Sukuk* Rating | 0.716 |
| Earnings Management | 0.742 |

Source: Output of SmartPLS 3.3.2.

The R square value is 0.716, which means that a 71.60% variation in *Sukuk* rating can be explained by FDQ, ABRs, and earnings management. Meanwhile, 74.20% of earnings management can be explained by FDQ and ABRs. The R square for *Sukuk* rating and earnings management are high enough as more significant than 0.670 Ghozali (2014).

### 4.6. Mediating Test Results (Intervening)

The intervening earnings management variable (which is the research gap of this research) is expected to mediate the relationship between the exogenous FDQ and ABRs variables on the endogenous *Sukuk* rating variable. This research uses the Sobel test to use a path analysis model between FDQ, ABRs, and the *Sukuk* rating. The results are presented in Figures 4 and 5. The Sobel test indicates FDQ, earnings management, and the *Sukuk* rating. It shows that the Sobel t statistic is greater than +2.109, and the test probability value is lower than 5% ($p < 0.05$). Thus, earnings management can mediate FDQ to *Sukuk* rating. However, the result also displays (see Figure 5) the relationship among ABRs, earnings management, and the *Sukuk* rating. It can be summed up that earnings management cannot be as mediating as the Sobel t statistic because it is lower than +2.109, and the test probability value is less than 5% ($p < 0.05$). Thus, earnings management cannot mediate ABRs to the *Sukuk* rating.

Please enter the necessary parameter values, and then click 'Calculate'.

A: -0.887
B: -0.230
SE$_A$: 0.191
SE$_B$: 0.079

Calculate!

Sobel test statistic: 2.46672750
One-tailed probability: 0.00681770
Two-tailed probability: 0.01363541

▸ Related Resources
x² Formulas   References   ⇌ Related Calculators   Q Search

**Figure 4.** Sobel Test Result of FDQ → Earnings Management → *Sukuk* Rating. Source: Sobel Test Output, available online: https://www.danielsoper.com/statcalc/calculator.aspx?id=31, accessed on 5 January 2022.

Please enter the necessary parameter values, and then click 'Calculate'.

A: 0.509
B: -0.230
SE$_A$: 0.297
SE$_B$: 0.079

Calculate!

Sobel test statistic: -1.47691617
One-tailed probability: 0.06984905
Two-tailed probability: 0.13969811

▸ Related Resources
x² Formulas   References   ⇌ Related Calculators   Q Search

**Figure 5.** Sobel Test Result ABRs → Earnings Management → *Sukuk* Rating. Source: Sobel Test Output, available online: https://www.danielsoper.com/statcalc/calculator.aspx?id=31, accessed on 5 January 2022.

EM cannot moderate the relationship between ABRs and *Sukuk* ratings. The data used in this study were only 3 years (2018–2020). In addition, the types of firms are heterogeneous with very different characteristics of variable data both in terms of fluctuations and the factors that influence it. During the research, global economic conditions, especially economic growth, were experiencing a downturn. However, the impact for each firm is different, i.e., the real estate industry experienced a decline.

In contrast, the industry sector based on ICT increased because of the decline in economics. In addition, if a significant level of (alpha) 10% is used, the result is substantial. It means EM can moderate the relationship between ABRs and the *Sukuk* rating. Therefore, the following research agenda can add data and indicators.

## 5. Discussion

### 5.1. The Effect of Financial Disclosure Quality (FDQ) on Sukuk Rating

Based on the first hypothesis test (H1), it can be concluded that the FDQ has a positive and significant effect on the *Sukuk* rating. The hypothesis formulated in this study that FDQ has a positive and significant effect on the *Sukuk* rating is accepted or proven. This finding is in line with the research results conducted by Qizam and Fong (2019). Jorion and Zhang (2007) and Qizam and Fong (2019) conducted a comparative study of the effect of FDQ on the bond *Sukuk* market in Indonesia. Besides that, they also performed a study of the criteria for rating agents

The financial statements disclosed by the firm are of a high quality if the level of relevance and reliability is high. Suppose the intrinsic and market values are more significant than the intrinsic and book values. In that case, the relevance of the financial statements disclosed by the firm is high. If the market value is greater than the book value,

the reliability is high. The firm's intrinsic value and market value are more significant than the intrinsic value and book value, indicating that the *PEFINDO* firm's complete and quality financial statement disclosures are well received by capital market players and raise the rating of the firm's *Sukuk*. Vice versa, if the financial statements disclosed by the firm are incomplete and of poor quality, the market will underestimate the firm's stock price, which will result in a low rating for the firm's *Sukuk*.

The results of this study are also in line with the publication of (Indonesian Rating *Sukuk* Agency) regarding the rating methodology. One of the statements, *PEFINDO*, can withdraw the published rating if the data and information provided by the *Sukuk* issuer or firm are insufficient or incomplete. In addition, the reliability of financial statements acts more as an indicator of the firm's FDQ. In other words, the more reliable the financial statements prepared by the firm are, the higher the *Sukuk* rating will be, and vice versa. However, the results are in line with the research of Qizam and Fong (2019), that the level of firm's reliability of the firm's financial statements is more decisive in the rating of *Sukuk* than the relevance of financial statements.

*5.2. The Effect of Accounting-Based Risks (ABRs) on Sukuk Rating*

The formulated hypothesis meets with the results of this research. The ABRs proxied by leverage have a negative and significant effect on the *Sukuk* rating. This finding is in line with the research result conducted by Elhaj et al. (2015), and Arundina et al. (2016).They suggested that financial leverage has a negative and significant effect on *Sukuk* ratings. However, this contradicts the research of Ningrum et al. (2019), Qizam and Fong (2019), and Al-Homsi et al. (2017) that financial leverage does not affect the *Sukuk* rating.

The leverage ratio is used to calculate the firm's debt level. How many of the firm's activities are financed by debt? The higher the firm's leverage ratio, the higher the firm's debt risk. Elhaj et al. (2017) noted that financial leverage measures the extent to which investors use loans or debt. The formula used to calculate the leverage ratio in this study is the debt-to-equity ratio (DER). The higher a firm depends on debt compared to its equity, the higher the risk of a debt default that will be faced and will lower the firm's *Sukuk* rating. The outcomes of this study support the theory that the firm's debt level is inversely proportional to the *Sukuk* rating. Therefore, the DER level of the industry is considered homogeneous. The DER for financial services, manufacturing, or trading companies is not heterogeneous.

*5.3. The Effect of Earnings Management on Sukuk Rating*

This study's third hypothesis (H3) is that earnings management has a negative and significant effect on the *Sukuk* rating; hence, the results are proven. The findings support Ulfa's (2019) research on the effect of financial ratios and non-financial factors on bond ratings. They concluded that earnings management has a negative effect on bond ratings. It is similar to the research by Ningrum et al. (2019) regarding the effect of financial ratios on *Sukuk* ratings with earnings management as an intervening variable.

Ruiz (2016) inferred that various motivations cause managers to carry out earnings management, namely contract motivation, compensation, loans, capital market motivation, and tax savings. According to positive accounting theory, as explained by Watts and Zimmerman (1986), the background of management in earnings management is the bonus plan hypothesis (increasing earnings to get bonuses), the debt covenant hypothesis (increasing earnings to maintain the reputation of debtors), and the political cost hypothesis (lower earnings to reduce taxes).

Based on the current theories, earnings management is a dangerous behavior by management in manipulating financial statements for the benefit of management. There are indications that manipulation of financial statements by management will lower the rating of the firm's *Sukuk*. The average firm that is the sample in this study performs earnings management practices by increasing earnings. Still, the average is close to 0; thus, the practice of earnings management by companies in this study is relatively low. Based on

the test results, the management behavior by performing earnings management negatively affects the firm's *Sukuk* rating.

These results are in line with the research of Kim et al. (2013). They found a relationship between earnings management and credit rating where the firm's credit rating assessment encourages management to carry out earnings management. However, the results of this study are not in line with the research of Yofrizal (2013), that earnings management does not affect the *Sukuk* rating.

### 5.4. The Effect of Financial Disclosure Quality (FDQ) on Earnings Management Sukuk

Based on the specific indirect effect test results or the mediating or intervening variable test, as shown in Figures 4 and 5, earnings management can mediate the relationship between FDQ and *Sukuk* ratings. However, earnings management cannot mediate the relationship between ABRs and *Sukuk* ratings.

The predictive function of earnings management (EM) variables can answer the inconsistency of previous research results as EM succeed to mediate the relationship between FDQ and *Sukuk*.

However, EM cannot mediate the effect between ABRs and the *Sukuk* rating. It means the portrayals of management's choice to make earnings adjustments are not driven by the presence of ABRs in the *Sukuk* rating. This finding is along with the research results conducted by Ningrum et al. (2019) and Ulfa (2019).

Moreover, EM cannot be moderating the relationship between ABRs and *Sukuk* Rating because of data reasons as described above in the section mediating test result. In addition, EM is a moral hazard behavior that can be done by the firm management in manipulating financial statements for management benefit. In accordance with positive accounting theory, as Watts and Zimmerman (1986) explained, the background for earnings management is to get more bonus (reward) and political cost strategy to increase earnings to attain good firm reputation.

The new model was resulted, it is because EM does not mediate ABRs on the *Sukuk* rating, the empirical model changes to the following Figure 6.

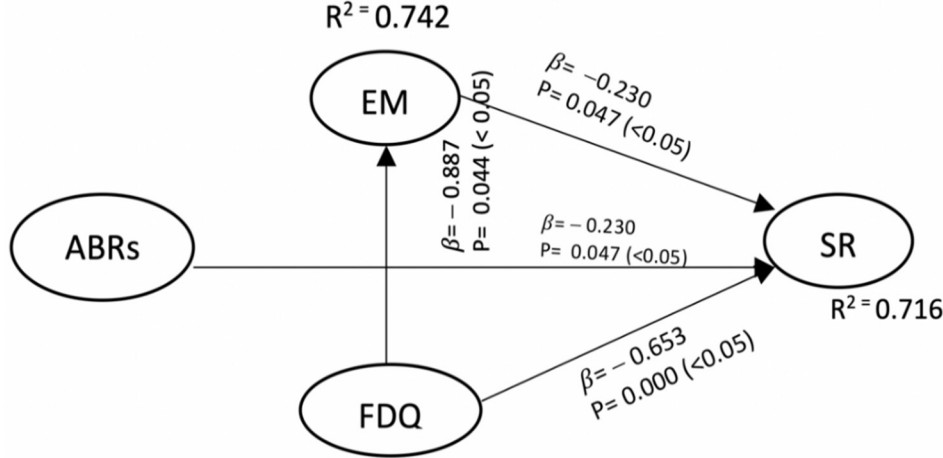

**Figure 6.** The Empirical Research Model.

### 5.5. Theoretical Implications

The scientific contributions theoretically this research are as follows: Firstly, the positive and significant effect between earnings management (EM) and *Sukuk* rating is crucial scientific contribution, namely earnings management variable strengthens the factors that influence the *Sukuk* rating. Secondly, EM succeeds to mediate FDQ on the *Sukuk* rating, which means contributing scientifically to other factors that influence the relationship between FDQ and the *Sukuk* rating.

Thirdly, EM does not mediate between ABRs and *Sukuk* ratings, this is expected to encourage further research by multiplying the data (time series) and the firms should be grouped by sectors. This inconsistency hopefully encourages new next research related *Sukuk* rating, and, fourthly, this study also found that using three grand theories, agency, asymmetry, and signaling theory, resulted in three main variables through the deduction process. The three variables are FDQ, ABRs, and earnings management and indicate they have a significant effect on the *Sukuk* rating. Accordingly, the three theories can be used for a new approach to the *Sukuk* rating study.

### 5.6. Managerial Implications

The managerial implications developed by this research are the results of this study are expected to contribute to the management of companies that issue *Sukuk*. They need to pay more attention to the completeness and quality of financial reports disclosed to the public, ABRs (leverage), and the practice of moral hazard in earnings management. Meanwhile, for investors, this research is expected to provide the importance of the completeness and quality of the financial statements from the companies issuing *Sukuk* to avoid the default risk. In addition, the factors of leverage and earnings management also need to be considered by investors.

## 6. Conclusions

The problem in this study was initiated by the inconsistency of the results in previous research, and phenomena gaps which is corporate *Sukuk* has increased in several year therefore this study employed several variables that is derived from three theory and resulted three variables FDQ, ABRs and EM.

In general, the result of this study in line with agency, asymmetry, and signaling theory. This theory assumed that all stakeholders (especially investor as agent and management as principal) should focus, interest, and pay more attention to their corporate inn order to achieve GCG, so that better firm performance (indicated by its *Sukuk* rating) can be acquired. This research approved that FDQ and ABRs have significant and positive effects on the *Sukuk* rating, while earnings management that is misleading or a bad moral hazard has negative effects on the *Sukuk* rating. This result means giving the signal to firms management to maintain and increase the trust, transference, accountability, and credibility as habit in the corporate culture.

In the context of *Sukuk* investments as part of Sharia bonds, investors may consider looking at the completeness and quality of the company's financial statements disclosed to the public compared to measuring earnings management practices and company leverage. Moreover, FDQ represented by the proxy of relevance and reliability of financial statements has a positive and significant effect on *Sukuk* rating.

Moreover, ABRs represented by the leverage ratio proxy and calculated by the debt-to-equity ratio formula have a keen and significant effect on the *Sukuk* rating. Earnings management calculated by the modified Jones model affects the *Sukuk* rating. FDQ, represented by the proxy of relevance and reliability of financial statements, has a negative and significant effect on earnings management. In addition, ABRs affect earnings management. Earnings management cannot correctly mediate the relationship between ABRs and *Sukuk* ratings. However, it can mediate the relationship between FDQ and *Sukuk* ratings. The relationship theory used, namely agency and asymmetry theories, including signaling theory, have an essential and significant role in developing the *Sukuk* rating. All variables derived from the relationship theory significantly affect the *Sukuk* rating. The next study could add another approach related to investors and principals on the *Sukuk* investment.

### 6.1. Research Limitations

This research is still far from perfect. The research object in this study is very varied and heterogeneous. It is homogeneous, consisting of various industries (manufacturing, finance, services, etc.) with different characteristics, but they are treated the same. Hence,

the research results can be biased. The FDQ variable is calculated using the market value of the firm's stock with a limited period of 3 years because the data were not available online before 2018. The number of samples in this study has not reached 1000. Thus, it is not satisfactory to generalize the conclusions.

*6.2. Suggestions for Future Research*

Based on the limitations of the research and analysis of the tests, several future research plans are the result of increasing *Sukuk* issuance from year to year and can be interpreted as expanding the firm's preference to raise business capital from instrument debt in the form of *Sukuk*. It needs to be balanced with research on *Sukuk* itself to contribute both theoretically and managerially, especially for investors in choosing investment instruments to avoid the risk of default. Based on the study results, we know that the heterogeneity of the type of industry at the firm affects the hypothesis testing. Hence, it is expected to group the companies for the following research based on the type of industry. Furthermore, it is necessary to explore data to start in 2015, like other countries, and develop comparative studies with other countries such as Malaysia, Bahrain, and the United Arab Emirates.

Future research can add other variables that have not been examined in this study, for example, the *Sukuk* structure stated in the rating methodology by *PEFINDO* (Pefindo 2018). There are 2 (two) types of *Sukuk* schemes, namely asset-based *Sukuk* and asset-backed *Sukuk*. In addition, there are also other variables related to guarantees for *Sukuk*, according to PEFINDO publications. However, the existence of warranties or guarantees does not significantly affect the possibility of default on bonds/*Sukuk*. Even so, the guarantees can immediately recover from the occurrence of defaults with a claim mechanism. Further research is also advised to try other methods of measuring earnings management, such as the Dechow et al. (2011) measurement model or real earnings management (REM) calculations instead of the discretionary accrual method. Based on the relationship theory derived from the agency and asymmetry theories, the researchers propose expanding the number of variables in a more detailed way in future research.

**Author Contributions:** Conceptualized and investigated, B.S.; methodology, B.S.; resources, B.S.; supervision, B.S.; data analysis, W.W.; writing methodology, W.W.; data curation, M.T.A.; writing draft preparation, M.T.A.; writing review and editing, K.A.; visualization and data curation, R.H.S. All authors have read and agreed to the published version of the manuscript.

**Funding:** This research received no external funding.

**Institutional Review Board Statement:** Not applicable.

**Informed Consent Statement:** Not applicable.

**Data Availability Statement:** Not applicable.

**Acknowledgments:** Olivia Fachrunnisa and Nunung Ghoniyah who give crucial suggestions related this issue.

**Conflicts of Interest:** The authors declare no conflict of interest.

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
