# Peer review of "The Determinant of Sukuk Rating: Agency Theory and Asymmetry Theory Perspectives"

_risks, doi:10.3390/risks10080150_

Round 1
Reviewer 1 Report
The paper is interesting and rather well-written. Anyway, there are remarks, which have to be addressed:
1) the authors write about the rating of bonds in Indonesia, alas, want to publish in an international journal. In order to raise the interest of international readers, please clarify if International Financial Accounting Standards (IRF) in Indonesia are applied,
2) if the system is applied just in Indonesia, what is the role of globalization;
3) how to make corporate social disclosures comparable?
Please extend part of the paper about nonexistent interests and the rating of bonds.
The paper can be published if accomplished.
Hajiyev, H.A. oglu, Stolyarova, M., Kalacheva, O., Malitskaya, V., Ivanova, Ya., Malysheva, L. (2021). International Financial Reporting Standards’ (IFRS) application peculiarities: a case study. Entrepreneurship and Sustainability Issues, 9(2), 255-267. http://doi.org/10.9770/jesi.2021.9.2(17)
Zeibote, Z., Volkova, T., Todorov, K. (2019). The impact of globalization on regional development and competitiveness: cases of selected regions, Insights into Regional Development, 1(1), 33-47. https://doi.org/10.9770/ird.2019.1.1(3)
Czaja-Cieszyńska, H., Kordela, D., Zyznarska-Dworczak, B. (2021). How to make corporate social disclosures comparable? Entrepreneurship and Sustainability Issues, 9(2), 268-288. http://doi.org/10.9770/jesi.2021.9.2(18)
Hermawan, A., Gunardi, A. (2019). Motivation for disclosure of corporate social responsibility: evidence from banking industry in Indonesia, Entrepreneurship and Sustainability Issues 6(3): 1097-1106. http://doi.org/10.9770/jesi.2019.6.3(17)
Author Response
Dear reviewer
Thank you very much for your very constructive comments and suggestion to our manuscript. We have revised the draft based on your suggestions
- we have incorporated discussion about IFRS in the introduction part (line 222)
- We have added the discussion on the role of globalization at line 228
- We have put additional discussion on corporate disclosures at line 455
- we have also extend discussion on about nonexistent interests and the rating of bonds

Reviewer 2 Report
The topic of current study 'The Determinant of Sukuk Rating: Agency Theory and Asymmetry Theory Perspectives' is of great significant. Sukuk or Islamic bonds hold a significant space in the capital markets around the world especially in the Muslim countries. Having said, it is of great importance to study the factors that can affect sukuk rating contextually. I have following suggestions for the authors.
The proxy used for ABR (Leverage ratio) needs more explanation. Why authors have used only this proxy for ABR? The argument must be supported by extant literature.
The manuscript contains unnecessary details in the introduction and hypotheses section. The authors must keep it concise and concrete and must update the version with current literature citations.
The paper should give clear descriptions of how the data are collected. How many firms are in the sample? How long is the sample period? The detailed variable definitions should also be given with appropriate proxies used especially for EM.
The authors must give scientific explanation to why EM does not mediate the proposed relationship. Is this due to data or context or both?
The presentation and language of the paper must be improved. I found repetition in the presentation of same concept at different points. A careful consideration should be given to the sentence structure, unnecessary explanations must be removed before updating the manuscript further.
Author Response
Dear Reviewer
Thank you for your very constructive comments and suggestions. We have revised our manuscript base on your suggestion.
1) we have added discussion on ABR measurement, please see explanation bellow the Table 2.
2) We have put additional description on the data to make it clearer, see research methods section
3) We have added explanation regrading the results of EM, see discussion section start from line 873
4) Regarding the language, we have sent the manuscript to the professional English proof reader before resubmit to the editor

Reviewer 3 Report
I am glad to be able to review this article. My doctoral dissertation concerned the topic of the rating of insurance companies. I believe that ratings are a powerful tool to support investor decisions. However, its imprecision should be emphasized. This is what the authors of the article did.
Accurately formulated and verified research hypotheses allow us to assess the influence of size (such as FDQ, ABR, or EM) on the rating and the possibility of manipulating their size. It is also essential to emphasize the phenomenon of information asymmetry.
The text was prepared very correctly - it has a very clear structure. It is worth emphasizing that the references in the literature are an introduction to the research (literature review and indication of the research gap), and the results of the authors' analysis are related to other studies.
To better understand the context of the study, I suggest adding a paragraph about the Sukuk rating market. Please write if and which agencies have issued the ratings and what these values ​​are.
Author Response
Dear Reviewer
Thank you very much for your constructive comments and suggestion on our manuscript
1) we have emphasized the phenomenon of information asymmetry on the manuscript, see section 2.2 and 2.7
2) We have added a paragraph about the Sukuk rating market in section 2.5

Round 2
Reviewer 2 Report
The authors have revised the manuscript and have incorporated most of the suggestions and corrections.